# Multimeric RGD-Based Strategies for Selective Drug Delivery to Tumor Tissues

**DOI:** 10.3390/pharmaceutics15020525

**Published:** 2023-02-04

**Authors:** Jordan Cossu, Fabien Thoreau, Didier Boturyn

**Affiliations:** 1University Grenoble Alpes, CNRS, DCM UMR 5250, F-38000 Grenoble, France; 2University Poitiers, Inst Chim Milieux & Mat Poitiers IC2MP, UMR CNRS 7285, F-86073 Poitiers, France

**Keywords:** RGD peptide, αVβ3 integrin, peptide–drug conjugate, multimeric RGD

## Abstract

RGD peptides have received a lot of attention over the two last decades, in particular to improve tumor therapy through the targeting of the αVβ3 integrin receptor. This review focuses on the molecular design of multimeric RGD compounds, as well as the design of suitable linkers for drug delivery. Many examples of RGD–drug conjugates have been developed, and we show the importance of RGD constructs to enhance binding affinity to tumor cells, as well as their drug uptake. Further, we also highlight the use of RGD peptides as theranostic systems, promising tools offering dual modality, such as tumor diagnosis and therapy. In conclusion, we address the challenging issues, as well as ongoing and future development, in comparison with large molecules, such as monoclonal antibodies.

## 1. Introduction

The selective, targeted delivery of drugs to tumors is a major challenge for cancer treatment, especially when modulating the poor pharmacokinetics of free drugs. This strategy prevents harming healthy tissues, unlike traditional chemotherapies. The successful application of monoclonal antibodies (mAbs), especially in oncology [1], has stimulated the development of a new class of macromolecules, such as antibody–drug conjugates (ADCs) [2]. Even if these new therapies are beneficial to patients, the production of mAbs requires a very expensive biotechnological procedure [3]. An alternative is to chemically produce small molecules that can specifically recognize tumor cells. In addition to mAbs, peptide–drug conjugates (PDCs) with lengths of less than 20 amino acids have been found to improve tumor targeting through the recognition of proteins that are overexpressed on tumor cell surfaces (erbB2, VEGF receptor, αVβ3 integrin, etc.) [4]. The most extensive studies have been conducted on αVβ3 integrin, a transmembrane protein receptor overexpressed within the tumor microenvironment that recognizes the conserved minimal RGD sequence found in several extracellular matrix (ECM) proteins, such as fibronectin [5], vitronectin [6], and laminin [7]. It is worth noting that αVβ3 is highly expressed on tumors but at very low levels on other tissues. This major discovery has stimulated the design of small peptides that contain the ubiquitous RGD triad sequence to selectively address αVβ3 integrin [8,9]. It is well-known than the cyclization of peptides limits structural conformations and provides better stability toward protease [10], and as expected, cyclic RGD peptides were found to interact with higher affinity than linear ones, especially c(-RGDf-N(Me)V-) [11], which is marketed under the name cilengitide (Merck) (Figure 1). These RGD peptides and analogues were first used as inhibitors of the interaction of ECM with integrins. Nevertheless, the clinical studies, in particular for the treatment of glioblastomas, were unsuccessful [12]. The failure of these studies was not the end of RGD projects but an opportunity to design new RGD conjugate compounds by adding cargo, such as imaging agents for diagnostics or drugs for therapy. To date, RGD peptides combined with radionuclides have been intensively studied as radiotracers for tumor imaging [13], with ongoing clinical trials for the PET/CT imaging of several cancers.

In parallel, the development of RGD–drug conjugates has gained great interest with the discovery of numerous cleavable linkers to release the drug in its active form inside cells [14]. Based on these studies, dimerization and, in general, multimerization of cyclic RGD peptides were found to improve the biological results. The rationale for the design of multivalent RGD peptides derives from the wide diversity of natural processes involving multivalent ligand–receptor interactions [15]. In the early 2000s, several groups developed multivalent RGD compounds with improved affinities by grafting multiple copies of RGD peptides to polymers [16], particles [17,18], proteins [19], and peptides [20,21] (Figure 1). Additionally, it has been shown that multivalency does not only improve the binding strength of a multivalent RGD ligand to its αVβ3 receptor but it also improves receptor-mediated internalization of the bound entity and is a powerful carrier for drugs [21]. It is important to note that the endocytosis of RGD-derivative αVβ3 integrin complexes was described to occur through endocytosis [22], and therefore, a subsequent endosomal escape of the drug is often necessary, especially for biomolecules that are sensitive to acidic conditions and enzymatic degradation. In this review, we focus on original and recent results based on the design of multimeric RGD-based compounds, i.e., the different scaffolds and linkers, used for tumor drug delivery.
Figure 1Structures of RGD compounds: (**a**) cilengitide [11]; (**b**) RGD-containing polynorbornene polymer [16]; (**c**) RGD-functionalized vesicle [17]; (**d**) RGD-displaying antibody [19]; (**e**) tetrameric RGD-containing cyclodecapeptide [21]; (**f**) tetrameric RGD-containing polylysine [20].
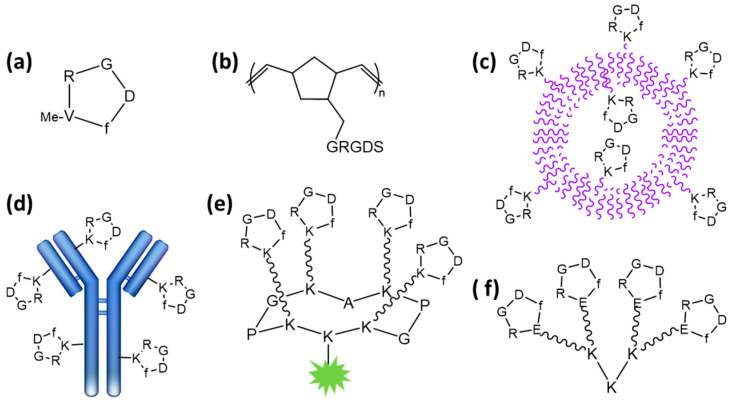



## 2. Design of Multimeric RGD Compounds for Drug Delivery

Numerous scaffolds are available to provide a multivalent presentation of ligands, which differ in their size, shape, and flexibility. These scaffolds can be classified into two main categories: low-molecular-weight scaffolds (with sizes < 7.5 kDa) and large macromolecules including polymers and nanoparticles (NPs). Depending on the scaffold structure, multivalent ligands can bind to their receptors through many possible mechanisms, including statistical rebinding, the chelate effect, and receptor clustering [23], which may influence their biological responses [24].

### 2.1. Using Low-Molecular-Weight Scaffolds

Over the past decades, small molecule–drug conjugates (SMDCs) have been greatly exploited for the selective delivery of anticancer cytotoxic agents [25], especially using peptides or peptidomimetics as scaffolds. These constructs provide pharmacokinetic properties that can be easily modulated with minimal immunogenic effects in comparison to large macromolecules. The interaction of SMDCs is mainly governed by several characteristics including valency, rigidity, and spatial organization. Well-known chemical synthesis methods are suitable for large-scale production, allowing accurate control of the payload-to-ligand ratio, and the versatility of the conjugation chemistry facilitates the modulation of in vitro and in vivo stability. In order to design such conjugates, suitable chemoselective ligations stable to degradative processes can been used, including amide and ether or thioether linkages, triazole rings, and thiol–maleimide and oxime ligation.

While RGD-containing low-molecular-weight compounds have been highly developed for imaging [26,27] and to design biomaterials [28], few of them have been developed for therapy. A pioneering study is certainly the use of a dimeric RGD compound for the delivery of paclitaxel (PTX) (Figure 2a) [29]. The results showed improved tumor specificity and cytotoxic effect of paclitaxel in an orthotopic breast cancer xenograft model, resulting in lower systemic doses to obtain antitumor efficacy. To assess the effect of conjugation, ^125^I-RGD and the conjugate were injected intravenously into tumor-bearing female athymic nude mice. Biodistribution studies confirmed that the PTX−RGD conjugate uptake was receptor-specific and comparable to the RGD uptake. More recently, Becker et al. developed other dimeric RGD compounds that were conjugated to Pt(IV)-based prodrugs via thiol–maleimide ligation (Figure 2b) [30]. Both fragments were connected by a flexible, Y-shaped core composed of a peptide chain with a terminal cysteine residue involved in the ligation with the Pt complex, a fluorescent label such as biotin or Cy5, and inert PEG27 to confer hydrophilicity and flexibility to the conjugate. These compounds showed interesting binding affinity in the nanomolar range, as well as selective and rapid internalization in colon carcinoma cells. The access of a tetravalent RGD-containing scaffold has been previously described using a cyclodecapeptide scaffold named RAFT (regioselectively addressable functionalized template) that exhibits suitable properties for drug delivery to tumors (Figure 1e and Figure 2c) [21,31]. A tetravalent RGD compound was shown to exhibit a 10-fold higher binding affinity toward the αVβ3 receptor compared to a monomeric analogue [32].

Further studies showed that a tetrameric RGD-modified compound was efficiently internalized through the clathrin-mediated endocytic pathway [22] and, thus, a good candidate for selective drug delivery. This construct has been exploited for the delivery of cytotoxic peptides able to destabilize mitochondria, therefore triggering cell apoptosis [33,34]. Interestingly, an intravenous injection of this cargo into humanized mice carrying human melanoma tumors showed tumor growth inhibition. Recently, this compound was also able to deliver a highly active cryptophycin derivative, showing impressive potency in M21 melanoma cells [35]. On the basis of these results, this multimeric system can improve selective, tumor-targeted drug delivery, providing a rationale for its future exploitation for therapeutic applications.

In parallel to classical RGD peptides, there is great interest in the development of RGD peptidomimetics and their multimeric versions. In this context, Manzoni et al. developed a panel of multimeric aza-bicycloalkane-RGD cyclopeptide (c(AbaRGD)) unit conjugates (Figure 2d) [36,37]. Glutamic acid dendrons (1–3 Glu) functionalized by robust amide or triazolyl bridges and a hydrolyzable ester bond allow the grafting of multiple respective c(AbaRGD) units and PTX. Biological assays have shown an enhanced affinity of tetrameric constructs in comparison to monomeric ones. An in vivo evaluation resulted in efficient tumor growth inhibition in a carcinoma model similar to that of PTX alone but with a favorable toxicity profile. More recently, another peptidomimetic construct, 4-aminoproline-based cyclotetrapeptide c(AmpRGD), was used to deliver the antiangiogenic kinase inhibitor sunitinib, which was clinically approved for several cancers (Figure 2e) [38]. Comparisons of mono- and dimeric c(AmpRGD) have shown better internalization of the dimeric compounds in human melanoma cells and a relevant antiangiogenic effect in mice. Similarly, Gennari et al. proposed a multivalent presentation of their previously developed c(DKP-RGD) (Figure 2f) [39] connected to paclitaxel through a lysosomally cleavable Val-Ala dipeptide and a self-immolative spacer, PABC-*N,N’*-dimethylethylenediamine. The results confirmed that multivalency improved the affinity of c(DKP-RGD)_n_–PTX conjugates. They revealed that dimerization and trimerization were beneficial regarding the vitronectin-binding inhibition when compared to a monomer. Interestingly, a plateau was reached as soon as three copies of RGD ligands were displayed. A reasonable explanation might be that the multivalent benefit on affinity was counterbalanced by an improved steric effect and reduced flexibility. Unfortunately, the absence of an internalization study prevented establishing the influence of multivalency on the fate of the multimeric PTX–c(DKP-RGD)_n_ conjugates. No in vitro nor in vivo cytotoxicity evaluations were reported.
Figure 2Structures of multimeric RGD compounds: (**a**) c(RGD)–paclitaxel compound [29]; (**b**) c(RGD) Y-shaped peptide core [30]; (**c**) tetrameric c(RGD)-containing cyclodecapeptide [31]; (**d**) c(AbaRGD)–glutamic acid dendrons [36]; (**e**) c(AmpRGD)–triazole, –ether, –amide linkages [38]; (**f**) c(DKP-RGD)–paclitaxel platforms [39].
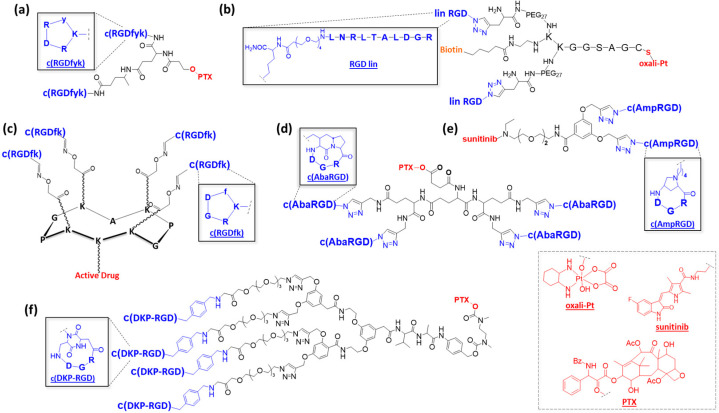


Altogether, multimeric RGD-containing compounds are highly effective for the delivery of potent anticancer agents. However, to date they are mostly exploited for imaging and diagnosis, as polymers and nanoparticles (NPs) can carry superior active drug payloads.

### 2.2. Using Polymers and Nanoparticles

To prepare multimeric RGD compounds for drug delivery, large nanostructures have been designed with a wide variety of sizes, shapes, and chemistries. Their interaction with the target is then governed by their structural and physicochemical properties, such as size, shape, nature of the scaffold, density, and accessibility to ligands [40,41,42]. These parameters have a great impact on biodistribution, cellular uptake, and toxicity. For instance, the best gold NP uptake into mammalian cells has been found using spherical particles of 50 nm [42,43]. The generation of nanomaterials functionalized with basic surface chemistries have been essentially developed to circumvent drug biocompatibility and toxicity. In this context, liposomes have proved to be efficient in drug delivery, as they modify the pharmacokinetics and biodistribution of the cytostatic agent, reducing the exposure of healthy tissues, such as the encapsulation of doxorubicin (Doxil), which received marketing authorization for several solid tumors [44]. Due to leaky blood vessels with pores varying from 100 nm to 2 μm in the tumor microenvironment, such nanocarriers are able to reach and to passively accumulate in tumors, a phenomenon named the EPR (employment of enhanced permeability and retention) effect [45].

Unlike low-molecular-weight scaffolds, the great advantage of carriers in the nanometer-sized range (polymers, organic and inorganic nanoparticles, liposomes) is undoubtedly the possibility to deliver large amounts of drugs. To improve tumor accumulation into cancerous cells or when the EPR effect is lacking, a new generation of NPs displaying targeting elements has been considered.

In the context of larger scaffolds, the multivalent presentation of ligands is generally obtained by the grafting of multiple copies of monovalent ligands on polymers or at the surfaces of NPs. Depending on the cargo, different polymers can be used. For nucleic acid delivery, due to their very poor intracellular uptake and limited blood stability, self-assembled NPs were designed using an RGD–polyethyleneimine (PEI) conjugate and SiRNA (Figure 3a) [46]. The cationic domain of the PEI bound to the polyanionic nucleic acid, generating stable nanoparticles (90–120 nm) using an amine:phosphate ratio of 2:1. These NPs were shown to exhibit specific SiRNA delivery after their intravenous systemic administration. Recently, a similar strategy was adopted using two different PEGylated polymers displaying the RGD tumor-homing peptide and a polycationic domain [47]. These polymers have been selected because, when internalized in endosome under acidic conditions, the rapid protonation of the cationic domains induces endosomal swelling and rupture due to osmotic phenomena, efficient endosomal escape of the nucleic acids, and the following expected biological effect [46,47]. By using the same strategy, other polymers can be used for the selective delivery of chemotherapeutic drugs to reduce their undesirable side effects. As proof of concept, most of studies have relied on the selective delivery of doxorubicin (Dox), which is exploited routinely in cancer chemotherapy. It is well-known that Dox can produce serious heart troubles, and the tumor-targeted delivery of this drug is recommended to prevent its toxicity. Dox can be chemically conjugated to RGD-modified polymers providing micelles [48] or to RGD-modified PEGylated polyamidoamine (PAMAM) dendrimers [49]. This drug can be also directly loaded in NPs using the hydrophobic interactions of Dox and polymer cores [50,51]. These Dox–RGD-modified polymers show significantly longer circulation times than free Dox and, as expected, better tumor uptake. Moreover, the choice of an RGD peptide sequence, such as iRGD, seemed important to improve the drug uptake [52]. Whatever the carrier used for the drug delivery, the release of Dox occurs following the cell internalization in the endosome after polymer degradation under acidic or reducing conditions. Thus, for example, RGD-decorated nanogels showed a better therapeutic effect with few adverse effects compared to free Dox on U87-MG humanized glioblastoma-tumor-bearing nude mice [51]. Similarly, other cytotoxic drugs, such as PTX [53,54], cisplatin [55], raloxifene [56], and temozolomide [57], have also shown promising therapeutic effects when encapsulated in RGD-modified polymers. 

In addition to self-assembled polymers, liposomes are well-adapted to encapsulate a wide range of components and, consequently, RGD-modified liposomes have been designed (Figure 1c). For example, Dox-containing RGD-modified liposomes were shown to significantly improve the in vivo drug efficacy in a mouse-bearing model of pancreatic cancer, as an antitumor effect was observed using 1 mg/kg of liposome while 15 mg/kg of free Dox was required for similar efficacy [58]. The same trend was observed for the delivery of paclitaxel [59]. RGD-modified liposomes showed the greatest tumor growth inhibitory effect in mice bearing PC-3 tumors compared to free PTX and PTX loaded in unmodified liposomes. 

In addition to organic-based NPs, different materials, such as silica, gold, iron oxides, and carbon oxides, have also been exploited to design NPs, and some are under evaluation in oncology clinical trials. They can combine drug and material properties in a synergetic effect. In this regard, mesoporous silica nanoparticles (MSNs) have interesting advantages, such as stable structure, large and well-defined surface area, and tunable pore sizes, which are advantageous for drug loading. In vitro results have shown that RGD-modified MSNs loaded with drugs, such as Dox and chlorambucil, enhanced cell growth inhibition efficiency in several cancerous cells [60,61]. Interestingly, MSNs can be loaded by several components to obtain a synergistic effect. Zhang et al. designed RGD-modified MSNs to combine chemo- and photothermal tumor therapy, respectively, by loading Dox and indocyanine green (IDG), with the latter able to generate hyperthermia under near-infrared light [62]. In vivo experiments in mice bearing 4T1 mammary carcinoma have shown higher tumor inhibition using RGD MSNs than a combination of free Dox and IDG, emphasizing the usefulness of the RGD tumor-cell-targeting motif. Furthermore, inorganic-material-based NPs are useful considering their intrinsic properties. In this context, RGD NPs consisting of reduced graphene oxide [63] and gold [64] have been used for their photothermal capacity. These works have showed significant biological effects in several cancer cell lines and in a chick embryo tumor xenograft, respectively. Superparamagnetic iron oxides (SPIOs) have also been used to design RGD NPs for their diagnostic and superparamagnetic properties in combination with chemotherapeutic drugs [65,66] for theranostic approaches (see also Section 4).
Figure 3Structures of RGD NPs: (**a**) SiRNA-containing RGD–polyethyleneimine NP [46]; (**b**) RGD–Gold NP with PEG corona [67].
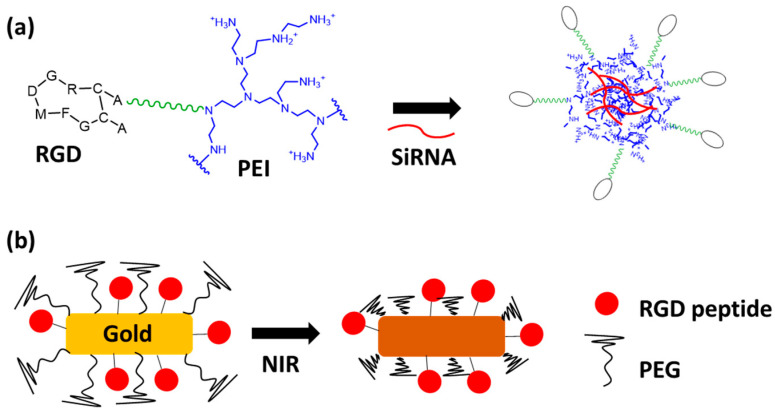



The latest generation of NPs relies upon smart-environment-responsive nanomaterials that should improve drug delivery. To control tumor selectivity, RGD NPs were designed using gold nanorods, and a temperature-sensitive polymer corona was used to increase passive accumulation at tumor sites via the EPR effect. Under NIR irradiation on the tumor site, the gold nanorods heated the surroundings, leading to the shrinkage of the polymer and the exposure of the RGD moieties and triggering NP tumor cell uptake (Figure 3b) [67]. Later, the same strategy was used with a PEG corona and a linker that was sensitive to matrix metalloproteinase-9 (MMP-9), leading first to passive accumulation. The PEG corona was removed by overexpressed MMP-9 in the tumor tissue, exposing RGD at the NP surface [68]. Biological studies in mice have shown higher tumor inhibition activity with minimal side effects. These results agree with previous findings, indicating that the EPR effect is favorable in cases of long-circulating nanomedicine formulations [69]. Another biological event can be used for the design of environment-responsive nanomaterials, such as reactive oxygen species (ROS), which are present in cancer cells in higher concentrations than in normal tissues. In this context, RGD NPs were prepared using an ROS-responsive anticancer mitoxantrone prodrug activated in tumor cells [70]. These smart, ROS-responsive NPs again showed significant inhibition of tumor cells in vivo.

As cancer cells have a different metabolism, the generation of smart NPs taking advantage of the cellular environment should continue to be exploited, especially to gain selectivity.

## 3. Use of Cleavable Linker for Multimeric RGD-Based Drug Delivery

The efficient delivery of a payload (drug or diagnostic agent) is obviously related to the targeting module employed to drive it to the desired site of action. However, the linker connecting them also has a crucial impact on the delivery outcome, making the payload, ligand, and linker selections highly intertwined. The mode of action of a payload often requires it to be located inside a cell to be enabled, while cell entry can be driven by various mechanisms, including passive diffusion or internalization mediated by the targeting module (e.g., peptide ligand, small molecule, antibody, etc.). The linker must then be wisely selected accordingly to multiple features—whether the targeted payload is required to be released, in a traceless fashion or not, in intra- or extracellular media in a specific cellular compartment, as well as the propensity of a ligand to trigger internalization or not.

For the purpose of this review, we focus on cleavable linkers. After a brief listing of the main families of cleavable linkers illustrated with RGD applications, a deeper emphasis is made on cleavable linkers employed for multimeric RGD-based drug delivery. The first large family of cleavable linkers is made of enzyme-sensitive linkers. Among them, a distinction can be made between linkers that encompass a peptide sequence as a cleavage site (peptidase substrate) and others, as depicted below.

### 3.1. General Overview: Enzymatically Cleavable Linkers

Numerous dipeptide sequences have been described as enzyme-sensitive, and many of them have been used in the context of RGD-targeted compounds. The most popular might be the Val-Cit sequence, known to be a substrate of cathepsin B that results in intra-cellular cleavage. It has often been used in combination with a PABC (para-aminobenzyl carbamate) spacer to allow traceless release, as in the commercialized brentuximab vedotin (Adcetris^®^) antibody–drug conjugate preferentially releasing MMAE to CD30+ cells [71], as well as to release a cryptophycin-55 glycinate drug from a monomeric RGD ligand [72]. Alternatively, Val-Cit-Gly-Pro has been used, where the Gly-Pro dipeptide was meant to decompose by diketopiperazine formation in order to release the drug in a traceless fashion [72,73]. It is worth noting that, despite the widespread of Val-Cit-PABC linkers, recent work indicated that cathepsin B could actually be useless for linker cleavage, instead carried out by other cysteine proteases [74]. If Val-Cit-PABC proves efficient for many triggered release applications, underlying mechanisms might need to be clarified. The Val-Ala dipeptide is another sequence particularly employed to design cleavable conjugates targeting integrin αVβ3. It has notably been introduced in linkers in combination with a PABC self-immolative spacer to trigger lysosomal traceless release of cryptophycin from RGD and isoDGR ligands [75], as well as traceless release of PTX, auristatin derivatives (MMAE or MMAF), and α-amanitin from DKP versions of cycloRGD and isoDGR, respectively [76,77,78]. Other di- and tetrapeptide sequences have been sporadically used as protease-cleavable linkers, such as Phe-Lys and Gly-Phe-Leu-Gly peptide sequences, all of them combined with a (PABC)-N,N′-dimethylethylenediamine self-immolative unit [76,79]. A way to circumvent moderate or inefficient internalization is to use an extra-cellular stimulus, such as an Asn-Pro-Val tripeptide sequence, a substrate of neutrophil-secreted elastase [80]. Very recently, a Gly-Pro-Ala tripeptide was used as a cleavable linker, to trigger a self-assembly that resulted in the tumoral accumulation of a so-called bioactivated in vivo assembly (BIVA) probe with a prolonged imaging window [81].

In the realm of enzyme-sensitive linkers, the disulfide bridge is particularly widespread. It is simple, quite stable in circulation, and an easily amenable unit that allows lysosomal cleavage and payload release upon disulfide reduction by the excess glutathione (GSH) present in subcellular compartments. The use of a linear RGD ligand connected to an aggregation-induced emission (AIE) probe and a gemcitabine unit via a GFLG linker (substrate of cathepsin B) and a disulfide linker (substrate of GSH), respectively, resulted in a dual-responsive αVβ3-targeted theranostic compound [82]. Other examples of disulfide linkers have permitted the release of camptothecin [83] or mitochondria-disruptive peptides [34]. It is important to note that matrix metalloproteinases and glycosidases have also been used to trigger extracellular payload release [84,85,86].

### 3.2. General Overview: Physically and Chemically Cleavable Linkers

The cleavage of a linker is not necessarily caused by an enzyme and can instead be triggered by physico-chemical stimuli. As a matter of fact, pioneering cleavable linkers are often meant to exploit the acidic media of subcellular compartments, such as lysosomes, to be cleaved and induce intracellular release. As such, chemical linkages, including ester, carbamate, (acyl)hydrazone, and acetal functions, have been employed to generate acido-labil linkers, with some of them combined with an RGD-targeting motif for the intracellular release of paclitaxel [29], ERK pathway inhibitor [87], Dox [88], and camptothecin [89], among others. Physical stimuli such as light can also be used to trigger the so-called uncaging phenomenon and trigger payload activation or release [90,91]. Interestingly, the click-and-release concept can be included in the panel of methods enabling the nonenzymatic cleavage of a linker [92,93,94]. However, to the best of our knowledge, no such application has been reported so far to release a payload from an RGD ligand. As a matter of fact, the click-and-release strategy might be considered as an extracellular trigger more suited for noninternalizing ligands.

### 3.3. Cleavable Linkers in Multimeric RGD-Based Drug Delivery

Multimeric presentations of RGD ligands can be combined with a cleavable linker to benefit from the high selectivity and affinity and effective internalization of a whole conjugate, as well as the triggered release of active compounds in integrin-αVβ3-positive cells. The first report of a multimeric RGD encompassing a cleavable linker was made by Chen et al. in 2005 (Figure 4, compound **1**) [84]. It consisted of an RGD dimer and a PTX unit connected by a hydrolyzable ester bond. The cytotoxicity of the resulting conjugate was compared to individual counterparts in MDA-MB-435 breast cancer cells (MTT assay). The E[c(RGDyK)]_2_–PTX conjugate revealed effective cytotoxicity (IC_50_ = 134 ± 28 nM), although lower than that of PTX alone (IC_50_ = 34 ± 5 nM). In a following study [95], the same authors evaluated the biodistribution and antitumor effect of the E[c(RGDyK)]_2_–PTX compound in an orthotopic MDA-MB-435 breast cancer model. The biodistribution of tritium radiolabeled versions of E[c(RGDyK)]_2_–PTX and PTX revealed a higher initial tumor exposure dose and prolonged tumor retention for E[c(RGDyK)]_2_–PTX than for free PTX. Ryppa et al. realized the in vitro and in vivo evaluations of a E[c(RGDfK)]_2_–PTX conjugate (Figure 4, compound **2**) incorporating a hydrolyzable ester linker, only differing from the above-described E[c(RGDyK)]_2_–PTX in the amino acid neighboring the RGD tripeptide (a phenylalanine rather than a tyrosine) [96]. They demonstrated convincing targeting specificity *in vitro* via a HUVEC cell adhesion assay and clear antiangiogenic properties of E[c(RGDfK)]_2_–PTX through three different *in vitro* assays on HUVEC cells (proliferation, migration, capillary-like tube formation). However, the short half-life of the ester bond resulted in the early release of PTX and similar activity on proliferation and migration for any of the E[c(RGDfK)]_2_–PTX, PTX, and control E[c(RADfK)]_2_–PTX samples when experiment duration exceeded a few hours. Unfortunately, the in vivo evaluation revealed no improved efficacy over free paclitaxel in the OVCAR-3 xenograft model. The authors suggested that more stable linkers or different drugs should be tested to improve the outcomes. An E-[c(RGDfK)_2_]-targeting ligand was then connected to Dox via a MMP2/MMP9 cleavable sequence (Figure 4, compound **3**) [97] sensitive to MMP2/MMP9 enzymes, which are overexpressed in tumor vasculature. The release of doxorubicin upon MMP2 cleavage was observed in an OVCAR-3 tumor, as opposed to the stable amide analogue. As for the previously described PTX–RGD dimers, the in vivo evaluation of the doxorubicin–RGD dimer did not reveal antitumor activity. The absence of in vivo activity might be partly due to the αVβ3-mediated internalization of E-[c(RGDfK)_2_]-Dox-_2_ being too fast for the extracellular MMP2/MMP9 to cleave the linker before it entered the cell. In 2010, Dal Pozzo et al. reported the synthesis and biological evaluation of a series of camptothecin–RGD conjugates of seven NMT-[c(RGDy-amF)]_2_ samples (Figure 4, compound **4**), including a cleavable Ala-Cit dipeptide sequence [98]. Their work demonstrated the crucial role of the linker between the targeting moiety and the payload, where fine tuning was essential to reach good equilibrium between length, solubility, and stability, as well as the release and targeting properties of the whole resulting compound. The two RGD dimeric compounds had similar binding affinities to isolated integrin αVβ3 and better cell adhesion inhibition capacity than monomeric RGD compounds. Overall, these results provide interesting information regarding cleavable linker tuning, but if the in vitro cytotoxicity was promising, the stability of the NMT–RGD_2_ compounds was moderate, and in vivo evaluation is required. In 2011, Polyak et al. reported the derivatization of a previously reported [c(RGDfK)_2_] dimer with a PEG chain and doxorubicin [99]. Noteworthily, an acid-sensitive (6-maleimidocaproyl)hydrazone linker was used to connect the RGD dimer to the drug (Figure 4, compound **5**), while a stable linker was used for the fluorescent dye. The free Dox internalized almost two-fold more than [c(RGDfK)_2_]-PEG-Dox. Concerning cytotoxic experiments on HUVEC and U87-MG cells, [c(RGDfK)_2_]-PEG-Dox showed a better IC_50_ (25 µM) than free Dox (40 µM). However, no in vivo evaluation of the anticancer activity of [c(RGDfK)_2_]-PEG-Dox has been reported so far. In parallel, a series of paclitaxel–cRGD monomers and a paclitaxel–cRGD dimer was reported [36]. Two different versions of the cyclic peptide were used, either AbaRGD (azabicycloalkane–RGD) or AmproRGD (aminoproline–RGD), and a scissile ester bond was used to connect the paclitaxel (PTX) to the RGD(s). As expected, the dimer PTX-[AmproRGD]_2_ demonstrated improved ability when compared to other monomers to inhibit biotinylated vitronectin binding to human isolated αVβ3 integrin (IC_50_ = 0.23 nM). Inhibition of tumor cell growth by PTX-[AmproRGD]_2_ was evaluated on IGROV-1 and IGROV-1/Pt1 cell lines (the second being a cisplatin-resistant subline). Satisfyingly, PTX-[AmproRGD]_2_ displayed similar activity to free PTX on both cell lines in the nanomolar range. This result indicated that PTX release was effective, but no evidence allowed the discrimination between extra- or intracellular release. Despite these encouraging results and without further justification, no in vivo evaluation of the dimer was reported. Other conjugates, including dimeric and tetrameric cRGD–paclitaxel conjugates based on AbaRGD ligands, were reported. A marked potency was obtained for all the compounds, with IC_50_ values in the nanomolar range, even lower than that of free PTX. The good in vitro results translated into encouraging in vivo outcomes: the PTX-[cRGD]_2_ dimer with short spacers was selected for the evaluation of in vivo antitumor activity and demonstrated similar tumor volume inhibition as free PTX. According to the authors, this result tends to indicate that the ester link of the dimer conjugate did not undergo premature PTX release. Recently, we reported the incorporation of a Val-Cit-PAB cleavable linker in a tetravalent cRGD construct in order to trigger the intracellular release of a cryptophycin-55 glycinate drug (Figure 4, compound **7**) [35]. A side-to-side comparison was realized on different cancer cell lines (U87 human glioblastoma and M21 human melanoma cells) with the free drug, the tetrameric RGD, the monomer conjugate c(RGDfK)-crypto, and tetrameric conjugate RAFT-c(RGDfK)_4_-crypto to determine cell viability. Both the tetrameric and monomeric RGD-cryptophycin conjugates induced a dose-dependent inhibition of cell growth in the U87 and M21 cell lines, with IC_50_ values in the nanomolar range. Interestingly, only the RAFT-c(RGDfK)_4_-crypto conjugate displayed the same toxicity as the free drug at the highest concentration (10 nM) on M21 cells, illustrating the improved internalization and integrin αVβ3 targeting enabled by the multimeric structure ensuring greater tumor selectivity. Additionally, the reduced activity of both RGD conjugates observed on M21-L underlined both the tumor selectivity and the stability offered by the ValCitPAB cleavable linker in the cell media (no premature release). Unfortunately, no in vivo evaluation of RAFT-c(RGDfK)_4_-crypto has been realized yet.

A lesson learned from the reported work is that cleavable linkers can be efficiently combined with multivalent RGD platforms to allow a triggered release inside or outside cells. However, we can notice that ester bonds are often used, while their stability in circulation is questionable, imparting the control of the release. Noteworthy, ester-cleavable linkers are nowadays rarely used in other delivery applications, such as antibody–drug conjugates [100]. Maybe this is one of the reasons why the in vivo assays of the reviewed multivalent RGD conjugates were almost all deceiving. This, as well as the fact that in vivo assays are not always realized despite interesting in vitro results, are possibly due to lack of facilities. Multivalent and cleavable linkers for controlled release are valuable tools, but it is likely that there is room for more research and new linker–multivalent ligand combinations to exploit their whole potential.

## 4. Multimeric RGD-Based Theranostic Systems

Theranostic strategies combining both diagnostic and therapeutic activities represent a new paradigm in the development of biotherapeutics, especially against cancer. Theranostic agents generally enable a primary diagnostic step to evaluate the tumor progression that also provides information on the potential tumor response to a subsequent treatment. Thus, theranostic agents tick the box of personalized medicine, which aims at selecting a treatment that a patient is most likely to respond to before executing it. As such, two types of theranostic agents can be distinguished: those encompassing two different effectors, typically an imaging agent and a cytotoxic payload, and those bearing an element that enables both detection and therapy, such as certain radioisotopes including ^64^Cu, ^177^Lu, and ^66^Ga, which combine both β particle emission for internal radiotherapy and γ particle emission for SPECT detection. Another possibility is combining radionucleotide pairs, such as ^90^Y/^86^Y [101] where either nucleotide can be inserted in the same complex, to exhibit complementary diagnostic or therapeutic activities. For the purpose of this review, we focus on multimeric RGD-based theranostic small molecules and only sporadically mention nano-objects.

### 4.1. Multimeric RGD Theranostic Systems Combining Multiple Effectors

There are actually few multimeric RGD theranostic systems that are reported to encompass both a diagnostic and a therapeutic effector in a small molecule conjugate. A smart way to achieve this is to incorporate a fluorophore into the linker (cleavable or not) that connects a RGD-targeting moiety to a drug. This was notably reported by Kim et al. in 2012 [102], followed by Gennari et al. in 2017 [83], who introduced a naphthalimide scaffold in their disulfide-containing linker that connected camptothecin and a c[RGDyK] or [DKP-RGD] ligand, respectively. In addition to CPT release, disulfide cleavage results in a redshift that enables the monitoring of compound localization, internalization, and payload release. Alternatively, Sewald et al. [103], introduced a carboxyfluorescein derivative in their cryptophycin-RGD compounds via a stable linker. If payload release cannot be monitored, the use of co-localization studies can confirm the conjugate internalization and cell localization. However, interesting examples actually concern monomeric RGD conjugates. To the best of our knowledge and quite surprisingly, no multimeric RGD construct has been reported to combine both a drug and a fluorescent probe. However, numerous larger constructs, such as nanoparticles [104], polymersomes [105], and quantum dots [106], have been reported that combine several copies of RGD ligands, drugs, and imaging agents.

### 4.2. Multimeric RGD Systems with a Single Theranostic Effector

Most of reported theranostic multimeric RGD systems actually include a radionuclide. This can probably be explained by design, historical, and facility reasons—having a single effector enabling both diagnosis and therapy is, de facto, simpler to design and synthesize than multi-effector compounds. In addition, nuclear medicine has been well-installed for many decades in hospitals and has proved efficacy, so the translation of multimeric RGD–radionucleotide systems from bench to bedside might appear more feasible. A series of multimeric RGD radionuclides based on the cyclodecapeptide named RAFT displaying four c[RGDfK] ligands and a DOTA complex enabled radiolabeling with various radionuclides. In 2015, DOTA-RAFT(c[-RGDfK-])_4_ was radiolabeled with either of two destructive β emitters, ^90^Y and ^177^Lu, which allowed internal targeted radiotherapy (Figure 5, compounds **9** and **10**) [107]. ^90^Y is a high-energy β emitter (E_max_ 2.28 MeV) with a half-life (T_1/2_ 2.7 days) compatible with that of peptides, as well as a long penetration range in tissues (R_max_ 11 mm). Therefore, the efficacy of ^90^Y-RAFT(c[-RGDfK-])_4_ was evaluated in mice bearing large αVβ3-positive tumors. Oppositely, ^177^Lu is a β emitter (78.7%, E_max_ 0.497 MeV) with a short penetration range in tissues (R_max_ 1.8 mm) and a longer half-life (T_1/2_ 6.7 days), making it more suited for irradiation of small tumors, and ^177^Lu-RAFT(c[-RGDfK-])_4_ was accordingly evaluated in mice bearing small αVβ3-positive tumors. Noteworthily, ^177^Lu is also a γ emitter (11%, E_max_ 0.208 MeV; 6.4%, E_max_ 0.113 MeV), rendering ^177^Lu-RAFT(c[-RGDfK-])_4_ a multivalent RGD theranostic compound. In vivo evaluation was realized on U-87 MG tumor-bearing mice. The biodistribution and organ uptake of ^90^Y-RAFT(c[-RGDfK-])_4_ were determined by ex vivo radioactivity measuring (with a γ-counter). SPECT/CT imaging after ^177^Lu-RAFT-RGD or ^177^Lu-RAFT-RAD (control group) injection revealed specific tumor accumulation with an excellent tumor/muscle ratio (about 10 at 1 h after injection). Overall, radionuclide therapy with either ^90^Y-RAFT(c[-RGDfK-])_4_ or ^177^Lu-RAFT(c[-RGDfK-])_4_ resulted in specific tumor growth inhibition and improved survival. ^90^Y-RAFT(c[-RGDfK-])_4_ treatment induced acute renal and bone marrow toxicities in mice, which could be reduced via injection fractionating. A RAFT(c[-RGDfK-])_4_ derivative was also functionalized with a cyclam chelator to enable the incorporation of ^64^Cu (Figure 5, compound **11**), allowing the PET imaging and therapy of αVβ3-positive tumors [108]. Mice with subcutaneous U87-MG glioblastoma xenografts received single administrations of 37 and 74 MBq of ^64^Cu-cyclam-RAFT(c[-RGDfK-])_4_ (37 MBq/nmol), a peptide control, or a vehicle solution, and the corresponding tumor growth was evaluated. Briefly, the results indicated that ^64^Cu-cyclam-RAFT(c[-RGDfK-])_4_ dose-dependently slowed down tumor growth, with mean tumor doses of 1.28 and 1.81 Gy from 37 and 74 MBq of ^64^Cu-cyclam-RAFT(c[-RGDfK-])_4_, respectively. Satisfyingly, this efficacy was accompanied with no apparent toxicity. Later, the same ^64^Cu-cyclam-RAFT(c[-RGDfK-])_4_ compound was combined with another theranostic compound, ^64^Cu-diacetyl-bis (N4-methylthiosemicarbazone) (^64^Cu-ATSM), meant to be a tracer of hypoxic metabolism [109]. Noteworthily, Cy5.5-RAFT(c[-RGDfK-])_4_ was also evaluated and demonstrated co-localization with ^64^Cu-cyclam-RAFT(c[-RGDfK-])_4_, thus appearing as a potential surrogate for the radioactive agent. On contrary, the intratumoral distributions of Cy5.5-RAFT(c[-RGDfK-])_4_ and ^64^Cu-ATSM were concomitant and nearly complementary, indicating that the combination of ^64^Cu-cyclam-RAFT(c[-RGDfK-])_4_ and ^64^Cu-ATSM could potentially result in a more uniform and optimized distribution of radioactivity. As a matter of fact, the combination of both ^64^Cu-cyclam-RAFT(c[-RGDfK-])_4_ and ^64^Cu-ATSM (18.5 MBq each) showed a sustained inhibitory effect against tumor growth and tumor cell proliferation, as well as prolonged survival of the mice, when compared to that of any of the single agent at the same total dose (37 MBq), while individual doses of 18.5 MBq induced no significant effects on tumor growth. Kidney and liver high uptakes accompanied these encouraging therapeutic effects, although without obvious adverse effects. More recently, we used a similar approach, combining ^64^Cu-cyclam-RAFT(c[-RGDfK-]_4_ and a Cy5.5-RAFT(c[-RGDfK-])_4_ fluorescent surrogate for the management of peritoneal metastasis in ovarian cancer. Intravenous or intraperitoneal administration of the multivalent RGD radionuclide was realized [110]. The intraperitoneal administration of ^64^Cu-cyclam-RAFT(c[-RGDfK-]_4_ resulted in high tumor penetration and enabled clear visualization of multiple peritoneal metastasis deposits according to PET analysis, while the biodistribution analysis demonstrated an inverse correlation between tumor uptake and tumor size. ^64^Cu-cyclam-RAFT(c[-RGDfK-]_4_) at a 148 MBq/0.357 nmol dose inhibited tumor cell proliferation and induced apoptosis with negligible toxicity. Overall, the panel of reported theranostic RAFT(c[-RGDfK-]_4_)–radionuclide conjugates proved efficient to combine both in vivo cancer diagnostic and therapy on mice in different types of cancers without toxicity. Hopefully these preclinical results are transferable to humans. As stated earlier, the ^177^Lu radionuclide enables both diagnosis (SPET) and therapy (internal radiotherapy). As early as 2012, a ^177^Lu-containing dimeric RGD conjugate was reported by Luna-Gutiérrez et al. [111]. However, it served, along with a monomer version, as a comparison point for the evaluation of gold nanoparticles encompassing multimeric copies of an RGD ligand and DOTA-complexed ^177^Lu. Still, the authors demonstrated interesting in vivo distributions of monomeric and dimeric ^177^Lu-RGD that were tumor-specific, displaying tumor-to-blood and tumor-to-muscle ratios that were higher than 3.5. Interestingly, the biological residence time and related radiation-absorbed dose for U87MG tumors were higher for the gold NP ^177^Lu-AuNP-c(RGDfK), but it also suffered from less specific tumor accumulation according to a blocking experiment. Not surprisingly, the dimeric conjugate ^177^Lu-RGD_2_ accumulated more in the kidneys and liver than the monomeric version but resulted in higher tumor accumulation as well (4.05 ± 0.04%ID/g for the dimeric conjugate versus 4.05 ± 0.04%ID/g for the monomeric conjugate expressed as the percentage of the injected dose per gram of tissue 1 h after injection). Unfortunately, no anticancer activity evaluation was realized. Shi et al. reported another dimeric RGD conjugate radiolabeled with ^177^Lu that they evaluated alone and in combination with an antiangiogenic therapy (using Endostar) on a U87MG tumor model [112]. Gamma imaging allowed tumor detection, and biodistribution studies revealed good tumor accumulation (notably, 6.03 ± 0.65%ID/g at 1 h and 3.55 ± 1.08 at 24 h after injection). Satisfyingly, tumor growth was significantly delayed by this targeted radiotherapy. The authors expected quick translation into clinical practice. Noteworthily, a single multimeric RGD compound bearing a free complex, such as DOTA or cyclam, can be considered for a theranostic approach if it is radiolabeled with a so-called radionucleotide “matched pair”, where a first radionuclide, typically a γ emitter, provides diagnostic capability (SPECT) to the resulting multimeric RGD conjugate, while radiolabeling with the other radionuclide, typically a β emitter, confers targeted internal radiotherapy capacity. It is the combination of a single targeted compound subjected to two different radiolabeling methods that enables a theranostic approach. An advantage of this approach is that biodistribution and tumor accumulation should be the same independently of the radionuclide, enabling good anticipation of the therapy response based on prior diagnostic evaluation. In addition, even though radionuclides such as ^177^Lu can be conveniently used for both diagnosis and therapy, such practice might induce a longer residence time of ^177^Lu and, thus, potential unnecessary exposure to radiation, as well as reduced image quality due to low abundance of γ rays. Using matched pairs avoid these limitations. Lee et al. reported two RGD dimers bearing an iminodiacetate (IDA) complex radiolabeled with either ^188^Re or ^99m^Tc (Figure 5, compounds **12** and **13**) [113]. The ex vivo biodistribution studies on U87MG-bearing nude mice revealed high tumor uptake with radioactivity levels of 12.3 ± 5.15%ID/g and 11.5 ± 1.41 at 1 h post-injection of ^99m^Tc-IDA-D-[c(RGDfK)_2_] and ^188^Re-IDA-D-[c(RGDfK)_2_], respectively, with quite high kidney uptake but low liver uptake. In vivo SPECT confirmed the interesting biodistribution results and the potential of ^99m^Tc- and ^188^Re-IDA-D-[c(RGDfK)_2_] to be used as a theranostic pair, but no anticancer activity evaluation was reported. Earlier, Liu et al. reported two DOTA-containing RGD dimers **14** and an RGD tetramer **15** (Figure 5) radiolabeled with the matched pair of ^111^In/^90^Y [114]. Biodistribution studies revealed that ^111^In-DOTA−3PRGD_2_ and ^111^In-DOTA−RGD_4_ exhibited similar tumor uptakes (e.g., 6.13 ± 0.82%ID/g vs. 6.43 ± 1.6%ID/g at 4 h post-injection respectively), but the uptake of ^111^In-DOTA−3PRGD_2_ in normal organs, including the liver and kidneys, was much lower than that of the tetramer, resulting in a significantly higher tumor/healthy tissue ratio and lower toxicity. As a consequence, the MTD of ^90^Y-DOTA−3PRGD_2_ in nude mice was more than 55.5 MBq, while that of ^90^Y-DOTA−RGD_4_ was less than 44.4 MBq. At the same dose, ^90^Y-DOTA−RGD_4_ and ^90^Y-DOTA−3PRGD_2_ showed similar and effective tumor growth inhibition, even though tumor volume kept increasing after injection. Altogether, these results demonstrate that good equilibrium in RGD multivalency must be found to optimize both tumor and nontumor accumulation to result in better efficacy and reduced toxicity. Interestingly in this case, the dimer exhibited similar antitumor activity as the tetramer but lower toxicity (higher MTD), making it a more suitable candidate for high-dose or multiple-dose regimens. Other multimeric RGD compounds have the potential to be used for theranostic applications, but the reported data have not included therapeutic evaluations [115,116,117].

Overall, the reported data confirm that multimeric RGD compounds are compatible with theranostic applications. However, it is likely that their potential has not been fully unleashed since several studies have been limited to their diagnostic or therapeutic applications instead of including both. In addition, multivalent presentations of RGD ligands have proved to be valuable to improve tumor selectivity and accumulation, but more can be the enemy of good, and it is clear that an optimal balance must be determined regarding the number of RGD copies to be used in order to keep an optimal tumor-to-healthy tissues ratio and improved therapeutic efficacy.

## 5. Conclusions and Future Perspectives

Active-targeting nanocarriers can significantly improve selective drug delivery to tumor tissues and, for this purpose, multimeric RGDs appear promising components, especially polymers and NPs that can carry larger drug payloads. Based on in vivo biological results, optimizing the loading of nanocarriers is then fundamental: the RGD ligand density should be optimized, as high densities of ligands could be more a problem than an asset [118]. This is due to steric hindrance, which can diminish affinity, but also to high cationic charges, which can be responsible for unspecific targeting, especially in the kidneys. In this context, linkers sensitive to the tumor or the intracellular microenvironment are appreciated to control the release of drugs. Additionally, it is important to note that the size and the shape of nanocarriers can strongly affect their fate in vivo [43], but little is known when comparing NPs and low-molecular-weight scaffolds. Overall, multivalent presentations of RGD ligands have been proved to have great potential for tumor targeting, but only a few approaches have been developed for therapy, principally due to a lack of specificity. Interestingly, the selective delivery of anticancer drugs to glioblastoma was recently demonstrated using an intranasal route [119]. This approach could pave the way for drug delivery to other types of brain diseases. In parallel, a new perspective of tumor-binding molecules, such as RGDs, is probably their combination with antibody-recruiting molecules to trigger a cytotoxic immune response [120].

## Figures and Tables

**Figure 4 pharmaceutics-15-00525-f004:**
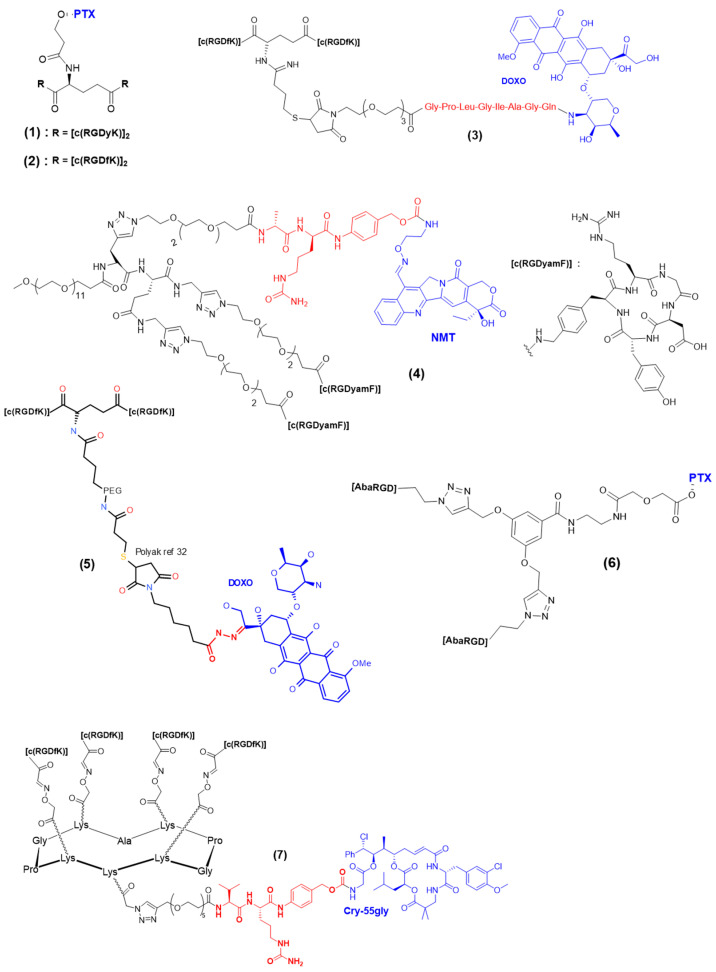
Multimeric RGD conjugates encompassing cleavable linkers.

**Figure 5 pharmaceutics-15-00525-f005:**
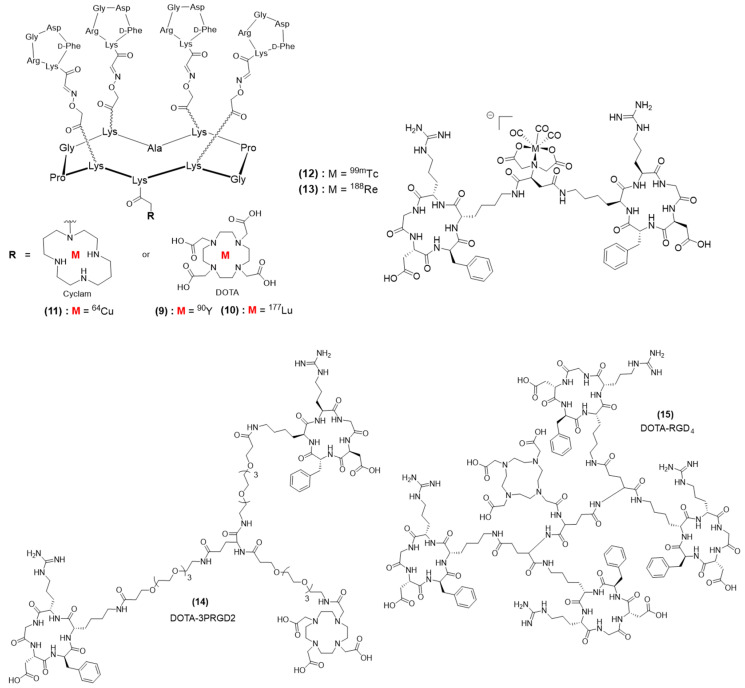
Theranostic multimeric RGD conjugates.

## Data Availability

Not applicable.

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
