# Peer review of "Multimeric RGD-Based Strategies for Selective Drug Delivery to Tumor Tissues"

_pharmaceutics, 2023, doi:10.3390/pharmaceutics15020525_

Round 1

Reviewer 1 Report

The authors summarized the recent advances in design of multimeric RGD-containing drug delivery system for tumor therapy and theranostics. Using the cyclic RGD mimic and various carrier scaffolds including small-molecule templates, antibodies, polymers and nanoparticles, RGD-based tumor-targeting strategies have been reported and introduced in this review.

Suggestion:

In the conclusion section, the authors discussed the challenges in this field, focusing on the structural optimization and tumor specificity enhancement as the future perspectives. Besides these issues, the in vivo pharmacokinetics and ADME properties of these RGD-conjugates are also very important for clinical applications. Especially, for the multimeric designs, the size of RGD-drug conjugates is usually large which might affect the in vivo pharmacokinetics. It will be helpful if such information could be discussed in the review.

Author Response

Point 1 : In the conclusion section, the authors discussed the challenges in this field, focusing on the structural optimization and tumor specificity enhancement as the future perspectives. Besides these issues, the in vivo pharmacokinetics and ADME properties of these RGD-conjugates are also very important for clinical applications. Especially, for the multimeric designs, the size of RGD-drug conjugates is usually large which might affect the in vivo pharmacokinetics. It will be helpful if such information could be discussed in the review.

Response to point 1:

We added in the conclusion part : "Additionally, it is important to note that the size and the shape of nanocarriers can strongly affect their fate in vivo [42], but little is known comparing NPs and low molecular weight scaffolds. "

Reviewer 2 Report

The manuscript wrote by Boturyn and coworkers nicely summarizes what is known about multimeric RGD-based strategy for delivering drugs and theranostic agent to cancer cells. To the best of my knowledge, the published literature is well covered, thus this review can represent a helpful tool for who is working on the topic.

I have just few comments for the authors:

1)      Row 41: Figure 1 in capital letter

2)     Figure 1 caption: please insert the letters “e)” and “f)

3)  Row 104-105: you mentioned here the Figure 1e, which is also repeated in Figure 2c. Please remove it from one Figure the identical structure or add here “and 2c”.

4)      Paragraph 2.1: among all the described compounds, are there examples that were tested also in vivo?

5)      Row 135: N,N should be in italic.

6)      Row 277: I think is missing the full name of PABC spacer, which may not be familiar for all readers.

7)      The sentence”…..cathepsin B could actually be dispensable for the linker cleavage, then carried out by other cysteine proteases” (row 284) is not clear. Please re-formulate it and add additional references (es. Mol. Cancer Ther. 15, 958–970 (2016)).

8)      Row 290-291: please add a-amanitin as toxin used in combination with Val-Ala linker (Beilstein J Org Chem. 2018; 14: 407–415)

9)      Row 293: The “or Val-Ala” should be delated since it was described few rows above. 

10)   Row 308: I would also describe the examples with MMPs and glycosidases sensitive cleavable linkers to give a complete overview of the used systems.

11)   Row 353: the MMP2 cleavage site is located between Gly4 and Ile5. Did they say nothing about the loss of potency of Dox due to the residual tetrapeptide sequence linked to the drug? It could be the main cause of the loss of activity, more than the fast internalization process as you specified.

12)   Conclusion and Future Perspectives: in row 598, you mentioned “lack of specificity”. I agree that the use of integrin-based ligands can lead to problems of specificity of the conjugates due to the expression of such receptors in healthy cell lines, though in lower expression. However, regarding the delivery of cytotoxic payloads, this concept is not really highlighted, but in my opinion it should, whenever is possible. For example, is the specificity (not the potency) increasing when the multimeric RGD-based drug-conjugate is used compared to the monomeric? Are there any data for to compare?

13)   Please be consistent with the “avb3” form throughout the paper.

Author Response

Point 1)      Row 41: Figure 1 in capital letter

Response: We have corrected

Point 2)     Figure 1 caption: please insert the letters “e)” and “f)

Response: We inserted e) and f) in the figure 1 caption.

Point 3)  Row 104-105: you mentioned here the Figure 1e, which is also repeated in Figure 2c. Please remove it from one Figure the identical structure or add here “and 2c”.

Response: We added "and 2c" row 104.

Point 4)      Paragraph 2.1: among all the described compounds, are there examples that were tested also in vivo?

Response: Yes in vivo experiments wre carried out for some compounds. We added in the text in vivo results :

row 94: "To assess the effect of conjugation, 125I-RGD and conjugate were injected intravenously into tumor-bearing female athymic nude mice. Biodistribution studies confirmed that PTX−RGD conjugate uptake was receptor-specific and comparable to the RGD up-take. "

row 117: "Interestingly, the intravenous injection of this cargo into humanized mice carrying human melanoma tumors have shown tumor growth inhibition."

Concerning row 130 and  row 136, in vivo results were already reported.

Point 5)      Row 135: N,N should be in italic.

Response: We have corrected this mistake.

Point 6)      Row 277: I think is missing the full name of PABC spacer, which may not be familiar for all readers.

Response: We added (para-aminobenzyl carbamate) just after PABC.

Point 7)      The sentence”…..cathepsin B could actually be dispensable for the linker cleavage, then carried out by other cysteine proteases” (row 284) is not clear. Please re-formulate it and add additional references (es. Mol. Cancer Ther. 15, 958–970 (2016)).

Response: we put "useless" instead of "dispensable" in the text.

Point 8)      Row 290-291: please add a-amanitin as toxin used in combination with Val-Ala linker (Beilstein J Org Chem. 2018; 14: 407–415)

Response: Row294, we added "auristatin derivatives (MMAE or MMAF) and α-amanitin " and a new reference 78. References were then updated.

Point 9)      Row 293: The “or Val-Ala” should be delated since it was described few rows above. 

Response: "Gly-Phe-Leu-Gly, or Val-Ala peptide " were replaced by "or Gly-Phe-Leu-Gly peptide ", rox 297

Point 10)   Row 308: I would also describe the examples with MMPs and glycosidases sensitive cleavable linkers to give a complete overview of the used systems.

Response: As these cleavable linkers were not generally used, we did not describe these studies including our work.

Point 11)   Row 353: the MMP2 cleavage site is located between Gly4 and Ile5. Did they say nothing about the loss of potency of Dox due to the residual tetrapeptide sequence linked to the drug? It could be the main cause of the loss of activity, more than the fast internalization process as you specified.

Response: I agree with the reviewer as the authors of ref 96 (Ryppa et al) should have designed a control Dox with flanking amino acids to compare biological activity with free Dox.

Point 12)   Conclusion and Future Perspectives: in row 598, you mentioned “lack of specificity”. I agree that the use of integrin-based ligands can lead to problems of specificity of the conjugates due to the expression of such receptors in healthy cell lines, though in lower expression. However, regarding the delivery of cytotoxic payloads, this concept is not really highlighted, but in my opinion it should, whenever is possible. For example, is the specificity (not the potency) increasing when the multimeric RGD-based drug-conjugate is used compared to the monomeric? Are there any data for to compare?

Response: It is well_known that increasing RGD unit increase the undesired kidney retention as mentioned row 597.

Point 13)   Please be consistent with the “avb3” form throughout the paper.

Response: we corrected in the text: αvβ3 were changed to αVβ3.

Reviewer 3 Report

This is a well-written review of multimeric RGD-based strategies for selective drug delivery to tumor tissues. I recommend it for publication after the points mentioned below are addressed.

1. In Figure 1, RGD-conjugated micelles should be added. One study (Biomacromolecules 2016, 17, 6, 2010–2018) related to this study should be included. 

2. The authors may have to add the copyright statements for the figures.

3. The authors should put the full name first when the abbreviation appears for the first time, such as RAFT.

4. Could the authors discuss how to optimize the number of the RGD groups on the nanoparticle?

Author Response

Point 1. In Figure 1, RGD-conjugated micelles should be added. One study (Biomacromolecules 2016, 17, 6, 2010–2018) related to this study should be included. 

Response: We added the reference [18] row 59. All references were then updated.

2. The authors may have to add the copyright statements for the figures.

Response: All figures were made by ourself.

3. The authors should put the full name first when the abbreviation appears for the first time, such as RAFT.

Response: we erased (selectively addressable functionalized template) in row 466

4. Could the authors discuss how to optimize the number of the RGD groups on the nanoparticle?

Response: this is a very interesting point but depending of the scaffold the optimization of the number of ligands will be different if it is a post functionalization for example we can mix RGD ligand with a non ligand compound. We did not focus this review on how design NPs.